# Incorporation of Zinc Oxide Nanoparticles Biosynthesized from *Epimedium brevicornum* Maxim. into PCL Nanofibers to Enhance Osteogenic Differentiation of Periodontal Ligament Stem Cells

**DOI:** 10.3390/ma18102295

**Published:** 2025-05-15

**Authors:** Kuei-Ping Hsieh, Parichart Naruphontjirakul, Jen-Hao Chen, Chih-Sheng Ko, Chi-Wei Lin, Wen-Ta Su

**Affiliations:** 1Department of Chemical Engineering and Biotechnology, National Taipei University of Technology, Taipei 106344, Taiwan; dog106344@gmail.com (K.-P.H.); cf1811@ntut.edu.tw (C.-W.L.); 2Biological Engineering Program, Faculty of Engineering, King Mongkut’s University of Technology Thonburi, Bangkok 10140, Thailand; smallwei2018@gmail.com; 3School of Dentistry, College of Dental Medicine, Kaohsiung Medical University, Kaohsiung 807378, Taiwan; tyhuang20181105@gmail.com; 4PhytoHealth Corporation, Taipei 105403, Taiwan; monkey106344@gmail.com

**Keywords:** green synthesis, zinc oxide nanoparticles, electrospinning, periodontal ligament stem cells, osteogenic differentiation

## Abstract

The optimal parameters for the microwave-assisted extraction of *Epimedium brevicornum* Maxim. were determined by using response surface methodology (RSM), increasing the extraction of flavonoids by 1.79 times. The resulting extract facilitated the green synthesis of zinc oxide nanoparticles (ZnONPs) with a wurtzite structure through a reaction with zinc nitrate. These ZnONPs were then incorporated into polycaprolactone (PCL) by using an electrospinning technique to produce nanofibers. The incorporation of ZnONPs resulted in an increase in Young’s modulus, biodegradation rate, and swelling ratio while decreasing the diameter and water contact angle of the nanofibers, thereby improving the hydrophilicity of PCL. ZnO demonstrates excellent biocompatibility with periodontal ligament stem cells (PDLSCs), increasing cell proliferation and enhancing alkaline phosphatase activity by 56.9% (*p* < 0.05). Additionally, mineralization deposition increased by 119% (*p* < 0.01) in the presence of 1% ZnO and showed a concentration-dependent response. After inducing PDLSC cultures with PCL–1% ZnO for 21 days, the protein expression levels of Runx2 and OCN increased by 50% (*p* < 0.05) and 30% (*p* < 0.001), respectively. Additionally, Col-1, Runx2, BSP, and OCN gene expression levels increased by 2.18, 1.88, 1.8, and 1.7 times, respectively. This study confirms that biosynthesized ZnONPs improve the physical properties of PCL nanofibers and effectively induce the osteogenic differentiation of PDLSCs.

## 1. Introduction

Zinc oxide is a II–VI semiconductor material with a wide band gap, light-emitting properties, piezoelectricity, and low cost. Zinc is ubiquitous in cells as a crucial component of various enzymes [1] and plays an extremely important role in human metabolism and protein and nucleic acid synthesis [2]. Zinc plays a crucial role in normal bone growth and bone homeostasis. The incorporation of zinc into biomaterials has been shown to enhance osteoblastic differentiation by promoting the expression of bone marker proteins such as alkaline phosphatase, type I collagen, and osteocalcin [3,4]. Zinc oxide nanoparticles (ZnONPs) are a prevalent type of zinc-containing nanomaterials renowned for their low cytotoxicity, biocompatibility, bioactivity, and chemical stability, garnering increasing attention in biological research. ZnONPs have been shown to accelerate bone growth and mineralization and exhibit selective toxicity towards bacteria and normal cells [5,6]. These properties endow ZnONPs with significant potential for orthopedic applications. ZnONPs can release Zn^2+^ ions as essential trace elements, which participate in various enzyme catalysis reactions and promote bone growth, mineralization, and formation [7].

Biological nanoparticle synthesis achieves non-toxic, low-contamination, and high-yield nanoparticles and is a green synthesis approach primarily using plant extracts or microbial fermentation broths [8]. These extracts contain various biomolecules, such as proteins, amino acids, carbohydrates, alkaloids, terpenoids, tannins, saponins, phenolic compounds, and vitamins. These molecules may participate in reducing, forming, and stabilizing metal nanoparticles [9,10]. This method involves a single synthesis step, operates under mild conditions, and reduces solvent and energy consumption to minimize environmental and human health impacts [11]. *Epimedium brevicornum* Maxim. (EbM), a traditional Chinese medicinal herb, exhibits various beneficial effects, such as enhancing immunity, anti-inflammatory and antitumor properties, anti-osteoporotic effects, anti-aging properties, endocrine regulation, and cardiovascular benefits [12]. Total flavonoids from EbM can influence osteoblast proliferation, differentiation, and mineralization, thereby promoting bone formation and reducing the number of osteoclasts, thereby decreasing their bone resorption function [13,14]. Therefore, extracts from EbM are highly suitable for synthesizing ZnONPs for orthopedic applications.

Polycaprolactone (PCL) is a hydrophobic polyester with biodegradability and biocompatibility and has been approved by the Food and Drug Administration for use in biomedical applications within the human body [15]. Electrospun nanofibers possess a porous structure, making them highly suitable for bone cell attachment and proliferation in bone defect repair [16]. Nanofibers are much smaller than human cells and can also form porous structures, which stimulate cell growth through environmental cues within the body [17,18].

Stem cells are primitive and unspecialized cells with self-renewal and differentiation potential, which allows them to either produce undifferentiated cells after division or differentiate and become specialized cells [19]. The periodontal ligament stem cell (PDLSC) is a type of stem cell first discovered in the periodontal ligament by Seo et al. [20] and successfully isolated and purified by Gay et al. in 2007 [21]. PDLSCs exhibit excellent differentiation potential and can effectively differentiate into osteogenic, chondrogenic, and adipogenic lineages, demonstrating the fundamental characteristics of mesenchymal stem cells [22,23]. Iwata et al. improved the purity of PDLSCs up to 96.7% of PDLSCs and demonstrated that they have better osteogenic ability than other MSCs under certain conditions [24]. Osteogenic differentiation refers to the biological process through which stem cells or osteoprogenitor cells undergo a series of molecular and cellular events that lead to their maturation into osteoblasts and ultimately osteocytes [25]. This process involves the sequential activation of specific signaling pathways and the expression of key osteogenic markers, such as Runx2, alkaline phosphatase (ALP), bone sialoprotein (BSP), and osteocalcin (OCN), which regulate the formation, mineralization, and maintenance of bone tissue.

This study aims to evaluate the effect of ZnONPs synthesized from *E. brevicornum* Maxim. extract (EbME) in promoting the osteoblastic differentiation of PDLSCs. ZnONPs were mixed with PCL to fabricate porous nanofibers, and the properties of the nanoparticles and nanofibers were analyzed. Cellular viability was measured via an MTT assay. Real-time polymerase chain reaction (RT-PCR), Western blot analysis, and Alizarin Red S staining were performed to determine matrix maturation and mineralization.

## 2. Materials and Methods

### 2.1. Isolation and Identification of PDLSCs

The steps for the isolation and purification of PDLSCs were performed as in our previous study [26]. The following standard operation procedure was approved by the Institutional Review Board of the Dental Clinic of Kaohsiung Medical University (KMUHIRB-SV(I)-20210047). The isolated cells were cultured in a medium composed of α-MEM (Gibco; Thermo Fisher Scientific, Waltham, MA, USA) with 10% fetal bovine serum, 100 mM L-ascorbic acid 2-phosphate (Sigma-Aldrich, St. Louis, MO, USA), and 1% antibiotic–antimycotic (Gibco; Thermo Fisher Scientific, Waltham, MA, USA) and incubated at 37 °C with 5% CO_2_. Subculturing was performed at 80% confluence for ordinary cultures, and the medium was changed every two days.

### 2.2. Preparation of Extracts of EbM Using Response Surface Methodology

EbM was purchased from a local Chinese medicine store and crushed into powder by using a grinder. The active compounds were extracted by using a microwave-assisted extraction device (MAS-II, SINEO, Shanghai, China). The mass of EbM powder was used fixed at 1 g, with time (min), ethanol concentration (%), temperature (℃), power (W), and solvent/solid (mL/g) being adjustable parameters. Initially, the effects of five individual variables on maximum flavonoid yield were evaluated by using a one-factor-at-a-time (OFAT) approach to identify the optimal range for each parameter. Subsequently, a two-level fractional factorial design was applied (Table 1), and the resulting data were subjected to regression analysis by using Design-Expert v13 software (Stat-Ease, Minneapolis, MN, USA). The results showed that ethanol concentration, temperature, power, and the solvent/solid ratio were statistically significant factors (*p* < 0.05) for maximizing the total flavonoid extract. Subsequently, the optimal extraction of total flavonoids was determined by using the steepest ascent path and central composite design. The EbME was initially concentrated under reduced pressure to remove most of the solvent and subsequently subjected to freeze-drying for storage.

The assay for total flavonoids was referenced from previous studies [27] and modified accordingly. First, 500 μL of EbME was mixed with 150 μL of 5% (*w*/*v*) sodium nitrite solution. The mixture was left to stand for 6 min before adding 150 μL of 10% (*w*/*v*) aluminum nitrate solution and then left to stand for another 6 min. Finally, 2 mL of 4% (*w*/*v*) sodium hydroxide solution was added, and the solution was made up to 5 mL with deionized water and left to stand for 60 min. The absorbance of the solution was measured at 415 nm by using a spectrophotometer (BioMate 3; Thermo Fisher Scientific, Waltham, MA, USA).

### 2.3. Synthesis and Characterization of ZnONPs

Next, 1 g of EbME was added to 100 mL of deionized water to prepare a 1% extraction solution. Then, 3 g of Zn(NO_3_)_2_·6H_2_O was added, and the solution was adjusted to a pH value of approximately 8 by using 1N NaOH. The mixture was stirred and reacted at 70 °C for 3 h. After that, the solution was centrifuged at 7500 rpm for 15 min, and the precipitate was collected after washing several times with deionized water and ethanol. The precipitate was dried in a 50 °C oven and then calcined in a 500 °C furnace for 2 h. The resulting product was collected for further analysis.

The physicochemical properties of ZnONPs were characterized by using various analytical techniques. UV–Vis absorbance spectroscopy (BK-UV1800, Biobase, Jinan, China) was used to confirm the characteristic absorption peak. The morphology, particle size, and elemental composition were examined by using scanning electron microscopy/energy-dispersive X-ray spectroscopy (SEM/EDS; JEOL JSM-6510, JEOL, Tokyo, Japan) and transmission electron microscopy (TEM; JEOL JJEM-2100F, JEOL, Tokyo, Japan). Selected area electron diffraction (SAED) and X-ray diffraction (XRD; X’Pert Powder, PANalytical, Malvern, UK) were performed to analyze the crystal structure. Additionally, Fourier transform infrared spectroscopy (FTIR; JT/IR-4600, Jasco, Tokyo, Japan) was employed to identify the functional groups.

### 2.4. Fabrication of PCL–ZnO Nanofibers and Physical Characterization

A 10% PCL solution was prepared by using hexafluoroisopropanol as the solvent. Different proportions of ZnONPs were added to prepare 0, 1, and 2% PCL–ZnO solutions, which were then sonicated for 30 min to ensure uniform suspension. These solutions were transferred into 10 mL syringes connected to an 18-gauge needle with Teflon tubing. Operating parameters were set as follows: 16 kV voltage, 20 cm collector distance, 0.015 mL/min flow rate, and 200 rpm collector rotation speed. Electrospinning (JSESC 10, Jyi-Goang CO., LTD., Taoyuan, Taiwan) was conducted at room temperature for 4 h. The resulting nanofibers were collected and stored in a humidity-controlled chamber for further experimentation.

The morphology and diameter of the PCL–ZnO nanofibers were examined by using scanning electron microscopy (SEM/EDS; JEOL JSM-6510, JEOL, Tokyo, Japan). Fiber diameter distribution was analyzed by randomly selecting 100 individual fibers and measuring their diameters by using image analysis software (ImageJ 1.50d, National Institutes of Health, Bethesda, MD, USA). Functional groups were identified through Fourier transform infrared spectroscopy (FTIR; JT/IR-4600, Jasco, Tokyo, Japan).

The swelling ratio of the PCL–ZnO nanofiber was calculated by using the following equation:Swelling ratio (%) = [(Ws − Wd)/Wd] × 100%(1)
where, Wd is the dry weight of the fiber, and Ws is the weight of the swollen fiber.

The mechanical strength of the PCL–ZnO nanofibers was determined by using a universal testing machine (FL-508M1, FALCO, Taipei, Taiwan). The biodegradability of the PCL–ZnO nanofibers was assessed by soaking them in phosphate-buffered solution (PBS) at 37 °C for 30 days; then, the residual weight of the fibers was obtained to calculate the degradation rate. The water contact angle was measured to determine the wettability of the fibers by using a contact angle measuring instrument (Model 100SB, Sindatek, Taipei, Taiwan). In addition, PCL–ZnO nanofibers were immersed in a PBS solution for 21 days, and the Zn ion concentration released into PBS was determined with an inductively coupled plasma–optical emission spectrometer (ICP-OES; Optima 8300, PerkinElmer, Inc., Waltham, MA, USA).

### 2.5. Cell Viability vs. ZnONP Treatment

PDLSCs with three to five passages were seeded into 96-well plates at 2 × 10^4^ cells/well with ordinary culture medium for one day. Then, the medium was replaced with fresh medium with or without 2 × 2 cm^2^ PCL–ZnO nanofibers. The cells were cultured further for 1, 2, or 3 days. Each well containing the proliferated cells was treated with 5 mg/mL MTT at 37 °C for 4 h. The cultivated medium was removed, and formazan was solubilized in DMSO. The metabolized MTT was measured based on the optical density at 570 nm by using a spectrophotometer (Multiskan FC, Thermo Fisher Scientific, Waltham, MA, USA).

### 2.6. Alkaline Phosphatase Activity Assay and Mineralization Analysis

Next, we seeded 2 × 10^4^ cells/well on the PCL–ZnO nanofibers with ordinary culture medium for 7 days, using pure PCL fiber as a control group. An alkaline phosphatase (ALP) assay was then performed, as in our previous study [28]. Finally, the absorbance was measured by using an ELISA reader (Multiskan FC, Thermo Fisher Scientific, Waltham, MA, USA) at a wavelength of 405 nm.

The calcification of differentiated cells on day 21 was quantified by using Alizarin Red S staining (Sigma-Aldrich, St. Louis, MO, USA), as in our previous study [26]. The solubilized preparation was obtained with 10% cetylpyridinium chloride. Images of stained cells were captured by using an optical microscope (CKX53; Olympus, Tokyo, Japan). The absorbance of the deposited calcium in the differentiated cells was measured at 570 nm (Multiskan FC, Thermo Fisher Scientific, Waltham, MA, USA). Based on the comprehensive analysis of cell viability, ALP activity testing, and mineralization analysis, PCL–1% ZnO was selected for subsequent experiments on bone differentiation protein and gene expression.

### 2.7. Reverse Transcription Polymerase Chain Reaction (RT-PCR)

PDLSCs were cultured on the PCL–ZnO nanofibers. The total RNA from cells was extracted by using TRIzol reagent (Ambion^®^, Life Technologies™, Carlsbad, CA, USA) for 10 min. RNA quantity was determined by using a NanoDrop 2000 spectrophotometer (Thermo Fisher Scientific, Waltham, MA, USA). cDNA was synthesized from 1000 ng of RNA by using the SOLI Script^®^ RT synthesis kit (Solis BioDyne, Tartu, Estonia) with a thermocycler (5 min at 65 °C, 30 min at 50 °C, 5 min at 85 °C; Veriti^®^ Thermal Cyclers, Applied Biosystems, San Francisco, CA, USA). Real-time polymerase chain reaction (RT-PCR) was performed by using HOT FIREPol^®^ EvaGreen^®^ qPCRSupermix 5x (Solis BioDyne, Tartu, Estonia) according to the manufacturer’s protocol. Briefly, an initial denaturation step was performed at 95 °C for 12 min, followed by 40 cycles of 95 °C for 15 s, 60 °C for 30 s, and 72 °C for 30 s with a Real-Time PCR System (QuantStudio™ 3, Thermo Fisher Scientific, Waltham, MA, USA). The relative fold change in gene expression was determined by using the 2^−∆∆Ct^ method. The PCR primers were as follows: Col-1 (forward: 5′-TGC TTG AAT GTG CTG ATG ACA GGG-3′; reverse: 5′-TCC CCT CAC CCT CCC AGT AT-3′); Runx2 (forward: 5′-TCC TGT AGA TCC GAG CAC CA-3′; reverse: 5′-CTG CTG CTG TTG TTG CTG TT-3′); BSP (forward: 5′-ATG GAG ATG GCG ATA GTT CG-3′; reverse: 5′-TCC ACT TCT GCT TCT TCG TCC-3′); OCN (forward: 5′-CTC TGC CTT AAA CAC ACA TTG-3′; reverse: 5′-TTC CCT TTG CCC ACC TC-3′); GAPDH (forward: 5′-ATG AGA AGT ATG ACA ACA GCC-3′; reverse: 5′-AGT CCT TCC ACG ATA CCA AA-3′).

### 2.8. Western Blotting

PDLSCs were seeded into 6-well culture plates at 2 × 10^5^ cells per well. The plates were then placed in a cell culture incubator and maintained at 37 °C with 5% CO_2_ for 24 h. After incubation, fresh culture medium with or without PCL–ZnO nanofibers was added to each well, and the cells were further cultured for 7, 14, and 21 days. Total cell protein was extracted by lysing the cells in lysis buffer, and protein concentrations were determined by using BCA protein assays at an absorbance of 595 nm (Multiskan FC, Thermo Fisher Scientific, Waltham, MA, USA). All samples were mixed with dye and heated in a water bath at 95 °C for 5 min. They were then added to the SDS-PAGE/electrophoresis tank along with Running Buffer and marker and were electrophoresed at 60 volts for 30 min, followed by 120 volts for 60 min. The PVDF membrane was placed over the SDS-PAGE gel and put into the transfer tank (Bio-Rad, Hercules, CA, USA) with Transfer Buffer. The transfer was carried out at 340 mA for 90 min at 4 °C, subsequently blocked with blocking buffer (5% skimmed milk powder in tris-buffered saline containing 0.1% Tween-20) for 1 h, and then probed with the primary antibody (Runx2, OCN, and β-actin, at 1:800; Abcam, Cambridge, UK) for 16 h at 4 °C. The blots were washed and incubated with secondary antibodies (goat anti-rabbit IgG H&L, 1:10,000; Sigma, St. Louis, MO, USA) on an orbital shaker for 1 h and then washed with 0.1% Tween-20 in tris-buffered saline. The proteins were visualized via chemiluminescence (LAS-4000; Fujifilm, Minato, Japan) by using the Amersham ECL (ThermoFisher Scientific, Waltham, MA, USA).

### 2.9. Statistical Analysis

All experiments were performed thrice for different samples. All data are expressed as means ± standard deviation (SD). Statistical analysis was conducted by using IBM SPSS Statistics 23 software, employing the least significant difference (LSD) post hoc multiple comparison one-way analysis of variance (ANOVA) test. A *p*-value less than 0.05 for the compared groups was considered statistically significant.

## 3. Results

### 3.1. Characterization of PDLSCs

The isolated and purified PDLSCs exhibited key mesenchymal stem cell markers, including CD166 (99.02%), CD90 (99.50%), CD73 (98.95%), CD105 (94.89%), and CD146 (76.64%). In contrast, the cells showed no expression of the hematopoietic markers CD34 (0%) and CD45 (1.08%).

### 3.2. Optimization of EbM Extract via RSM

The first-order Equation (2) (R^2^ = 0.9012) was solved through the regression analysis of a two-level factorial design for maximum total flavonoids.Total flavonoids = 67.60 + 4.97X_2_ + 5.23X_3_ − 5.12X_4_ + 2.91X_5_(2)

X_2_, X_3_, X_4_, and X_5_ are the coded values of the tested variables of ethanol concentration (%), temperature (°C), power (W), and solvent/solid ratio (mL/g), respectively. Due to the insignificant difference in extraction time (X_1_) and considering that the boiling point of alcohol at 78.2 °C might affect the accuracy of the model, the extraction time (X_1_) and temperature (X_3_) were fixed at 30 min and 70 °C, respectively. A steep path was designed (Table 2). The experimental results showed that step 2 achieved the highest total flavonoid yield. We then designed seventeen groups of experiments by using the central composite design (Table 3). The regression analysis results yielded the following optimized equation:Total flavonoids = 86.72 − 0.6200X_2_ − 0.6180X_4_ + 0.6509X_5_ − 1.83X_2_ X_4_−0.4537X_2_ X_5_ − 0.4837X_4_ X_5_ − 3.22X_2_^2^ − 3.10X_4_^2^ − 0.8404X_5_^2^(3)
where R^2^ = 0.9062, the coefficient of variation % = 2.17% < 10%, and *p*-value = 0.0072, indicating that the regression exhibited significant correlation (*p* < 0.05) and high reliability and accuracy.$$0.11\;*\;=\frac{1}{\sqrt{{X_{1}}^{2}+{X_{2}}^{2}+{X_{3}}^{2}+{X_{4}}^{2}+{X_{5}}^{2}}}$$

A three-dimensional response surface was plotted based on a statistically significant model to understand the interactions among the variables and the optimal quantities required. The highest point in Figure 1 denotes the maximal total flavonoid extract yield, which the RSM predicted would occur with the following parameter values: 30 min, 69.1% ethanol, 70 °C, 494.7 W, and 1114.66 mL/g solvent/solid ratio. The predicted yield was 86.92 mg/g quercetin. Under optimal conditions, the actual experimental yield was 86.84 mg/g quercetin, deviating from the predicted value by 0.09%.

### 3.3. Characterization of ZnONPs and PCL–ZnO Nanofibers

Figure 2A presents the UV–Vis absorption spectra of the prepared ZnONPs. The peak observed at 345 nm is characteristic of ZnONPs because they represent the intrinsic absorption of ZnO when electrons are promoted from the valence band to the conduction band of the semiconductor [29]. The crystal properties of ZnONPs were measured by XRD, and the reflection peaks were observed at the 2θ values of 31.78°, 34.43°, 36.25°, 47.56°, 56.64°, 62.88°, and 68.18° (Figure 2B), corresponding to the (100), (002), (101), (102), (110), (103), and (112) planes based on the standard wurtzite card (JCPDS No. 36-1451) for ZnONPs [30]. Based on the (100), (002), and (101) diffraction peaks, the average size of ZnONPs was calculated to be 11.3 ± 1.6 nm, confirming that the prepared ZnO is in the nanoscale. Figure 2C presents the FTIR spectra of ZnONPs and EbME in the wavenumber range of 400–4000 cm^−1^. The absorption peak at 3291 cm^−1^ is attributable to the –OH stretching vibrations, while the peak at 2920–2852 cm^−1^ arises from the asymmetric stretching of C–H bonds. In addition, the symmetric stretching of C=O at around 1706 cm^−1^, the stretching of C=C at 1608 cm^−1^, and the stretching of C–O at approximately 1040 cm^−1^ were observed [31]. The intense peak at 417 cm^−1^ corresponds to the ZnONPs [32]. With SEM, we observed the aggregation of ZnONPs into clusters with spherical morphology (Figure 2D), attributed to the spontaneous aggregation tendency arising from the small particle size, high surface energy, and high-temperature sintering. EDS analysis exhibited strong peaks corresponding to oxygen and zinc signals (Figure 2E), confirming the successful synthesis of ZnO. The TEM images confirmed that the prepared ZnONPs were predominantly spherical and square-shaped (Figure 2F), consistently with the SEM results. Particle size distribution was plotted by using Image J (Figure 2G), showing an average diameter of approximately 13 nm, which was consistent with that calculated with XRD analysis. With SAED, we analyzed the crystalline structure of the ZnONPs. As shown in Figure 2H, the results reveal different diffraction rings, indicating the polycrystalline nature of ZnONPs, corresponding to the hexagonal wurtzite structure observed in the XRD spectrum. The antibacterial results of ZnONPs against *Staphylococcus aureus* and *Escherichia coli* are shown in Figure 2I, showing the antibacterial ability of ZnONPs. The ZnO nanoparticles were analyzed by dynamic light scattering (DLS), revealing an average particle size of approximately 430 nm with a polydispersity index (PDI) of 0.596, which is notably larger than the 13 nm observed by TEM. This discrepancy is likely due to the aggregation of nanoparticles in suspension. The zeta potential of ZnO was measured at −23.73 mV (Figure 2J).

The SEM observation and fiber diameter analysis of PCL with or without ZnONPs are shown in Figure 3A. Smooth fibers with random alignment were observed in these samples. The diameter of pure PCL fibers was approximately 1118 nm, while those of PCL–1% ZnO and PCL–2% ZnO were 795nm and 728nm, respectively. This indicates that the addition of ZnONPs reduced the viscosity of the solution, resulting in a decrease in fiber diameter [33]. The FTIR analysis of the nanofibers is depicted in Figure 3B. Saturated C–H bonds are observed at 2945 cm^−1^ and 2865 cm^−1^, while C=O stretching vibrations are detected at 1723 cm^−1^. Additionally, C–C bond stretching is noted at 1294 cm^−1^, and asymmetric and symmetric C–O–C bond stretching is observed at 1241 cm^−1^ and 1166 cm^−1^, respectively [15]. The addition of ZnO does not significantly affect the absorption spectrum of the pure PCL nanofibers. The XRD analysis of the impact of ZnONPs on the crystalline nature of the PCL nanofibers (Figure 3C) reveals a sharp diffraction peak at 21.5° corresponding to the (110) plane and a relatively low-intensity peak at 23.5° corresponding to the (200) plane, consistent with the semi-crystalline nature of PCL [34,35]. Following the addition of ZnONPs, characteristic peaks of ZnO gradually emerge in the spectrum, with intensities increasing with the content of ZnONPs. However, no significant impact was observed on the intensity of the PCL diffraction peaks. Figure 3D illustrates the swelling capacity of PCL and PCL–ZnO nanofibers after immersion in PBS for one day. The results indicate that the addition of ZnONPs increases the degree of swelling of the nanofibers, with the degree of swelling progressively increasing with higher ZnONP content. Due to its hydrophobic nature, PCL exhibits a contact angle of 128.9 ± 0.5° (Figure 3E). Upon the addition of ZnONPs, there is a decrease in the contact angle by approximately 12–13%, indicating that the addition of ZnONPs can improve the hydrophilicity of PCL fibers. Figure 3F shows no significant difference in mechanical properties between PCL and PCL–1% ZnO. However, in the case of PCL–2% ZnO, there is a decrease in strain of approximately 82% and an increase in Young’s modulus by 6.9 times. After soaking the nanofibers in PBS for 30 days, degradation rates (%) of 17.0, 21.5, and 23.1 were observed, for PCL, PCL–1% ZnO, and PCL–2% ZnO, respectively (Figure 3G). Figure 3H indicates a burst release of Zn^2+^ between three and seven days for ZnO-containing nanofibers, followed by a slow release. The dissolution and release mechanism of zinc ions from the PCL nanofibers was investigated through kinetic modeling by using zero-order, first-order, Higuchi, Hixson–Crowell, and Korsmeyer–Peppas models. Among these, the release profile was best described by the first-order kinetic model, as it exhibited the highest coefficient of determination (R^2^), as shown in Appendix A). PCL and PCL–ZnO solutions exhibit shear thinning because the viscosity decreases as the shear rate increases (Figure I, left), and the shear stress vs. shear rate graph passes through the origin and shows a concave downward graph as the shear rate increases, confirming that PCL and PCL–ZnO are pseudoplastic fluids (Figure I, right).

### 3.4. PCL–ZnONP Nanofibers Induce Osteogenic Differentiation

Cytotoxicity and cell viability analysis confirmed that PCL fibers were not toxic to PDLSCs and even had a slight proliferative effect (Figure 4A,B). However, the addition of ZnONPs led to a decrease in cell survival, which decreased further with the increase in concentration, indicating that ZnO exhibits a certain degree of cytotoxicity. After seven days of culturing on PCL–ZnONP nanofibers, ALP activity was measured in PDLSCs, as shown in Figure 4C. The addition of ZnONPs significantly increased ALP activity by approximately 56.9% (*p* < 0.05) compared with pure PCL fibers, indicating the effective induction of osteogenic differentiation by ZnONPs. However, there was no statistically significant difference between 1% ZnO and 2% ZnO. After 21 days of culturing, differentiated cells on PCL–ZnONP nanofibers were stained with Alizarin Red S, as shown in Figure 4D. The absorbance values of both the PCL–1% ZnO and PCL–2% ZnO groups were significantly higher than that of the PCL group, with increases of 119% (*p* < 0.01) and 230% (*p* < 0.001), respectively. Additionally, the PCL–2% ZnO group showed an approximately 51% higher increase compared with the PCL–1% ZnO group, indicating effective mineralization induction by ZnONPs, with greater efficacy observed with higher concentrations.

### 3.5. Osteoblastic Gene and Protein Expression

After seven days of osteogenic induction culture of PDLSCs with PCL and PCL–ZnO nanofibers, the gene expression level of Col-1 in the PCL–ZnO group was 2.18-fold (*p* < 0.01) higher than that in the PCL group (Figure 5A). After 14 days of culture, the gene expression levels of Runx2 and BSP in the PCL–ZnO group were 1.88-fold (*p* < 0.01) and 1.80-fold (*p* < 0.01) higher, respectively, compared with the PCL group (Figure 5B,C). After 21 days of culture, the gene expression level of OCN in the PCL–ZnO group was 1.70-fold (*p* < 0.05) higher than that in the PCL group (Figure 5D). These results indicate that PCL–ZnO nanofibers have an inducing effect on osteogenic differentiation.

Figure 5E indicates that on days 7 and 14, the expression of the Runx2 protein in the PCL–ZnO group was consistently higher compared with the PCL group, with an approximate increase of 50% (*p* < 0.001). Similarly, in Figure 5F, the expression of the OCN protein showed a similar trend, with higher expression levels in the PCL–ZnO group on days 14 and 21. By day 21, the expression of OCN in the PCL–ZnO group was approximately 30% (*p* < 0.001) higher compared with the PCL group. Both protein expression results confirm the ability of ZnONPs to effectively induce osteogenic differentiation.

## 4. Discussion

Phytochemicals extracted from plants have been applied to produce metal oxide nanoparticles by green synthesis. In this study, ZnONPs were biosynthesized by using optimized extracts of EbM as reducing agents and stabilizers. The optimal parameters for EbM extraction were identified via RSM as follows: 30 min extraction time, 69.5% ethanol concentration, 70 °C extraction temperature, 500 W microwave power, and 115 mL/g solvent/solid ratio. Under these conditions, the total flavonoid content was measured as 86.84 ± 0.99 mg/g quercetin, which only deviates by 0.09% from the estimated value derived from the optimized equation in Equation (3). By designing a mathematical model with RSM, the yield of flavonoid extraction was improved by approximately 1.79 times.

Biosynthesis, primarily derived from plant or algal extracts and microbial cultured broth by reducing metal salts under mild conditions through bioactive components such as polyphenols, flavonoids, proteins, polysaccharides, saponins, and terpenoids [36,37] offers an environmentally friendly, safe, rapid, high-stability, clinically adaptable, and cost-effective approach [38]. Biosynthesized ZnONPs exhibit higher biocompatibility, antimicrobial activity, photocatalytic activity [39], plant growth promoting properties [40], and drug delivery and anticancer potential [41,42]. Zinc is an inorganic mineral that participates in various metabolic activities in the human body, including the activation of proteins involved in bone homeostasis, thereby accelerating bone growth and mineralization [43]. ZnONPs exhibit low toxicity, biocompatibility, bioactivity, and chemical stability, making them highly suitable for applications in orthopedic medicine [7].

In this study, ZnONPs synthesized by using EbME and zinc nitrate displayed spherical morphology with an average size of 13 nm, exhibiting a wurtzite crystal structure, and the characteristic peaks of ZnO were observed in both the UV–Vis spectra at 345 nm and FTIR analyses at 417 cm^−1^. These results are consistent with the characteristics of green synthesis of nano-ZnO using the leaf extract of *Azadirachta indica* (L.) [44]. Generally, ZnO synthesized from different plants and plant parts (leaves, roots, stems, flowers, or fruits) exhibits spherical or hexagonal structures with sizes ranging approximately from 6 to 60 nm [39]. Anil Kumar et al. [45] compared the functionality of ZnONPs synthesized via green and chemical synthesis and found that green-synthesized ZnONPs exhibit stronger photocatalytic activity and higher electrode reversibility.

Nanofibrous scaffolds fabricated by electrospinning biopolymers have been widely applied in regenerative medicine, and sub-micron-sized fibers exhibit a porous 3D structure, which closely resembles the extracellular matrix environment of natural cell growth [46]. This structure provides an excellent scaffold for cell attachment, proliferation, and differentiation during in vitro culture. The results of FTIR and XRD indicate that the addition of ZnONPs does not affect the structure of PCL nanofibers [47]. While the swelling ratio of the fibers slightly increases, wettability remains largely unchanged, indicating that the hydrophobic characteristic of PCL has been retained. In biomedical scaffolds, material degradation and drug release are of paramount importance. Figure 3G indicates that ZnONPs can enhance the degradation rate of PCL and increase ion release as the ZnONP content increases. The concentration of Zn^2+^ in PCL–2% ZnO reaches 2.39 mg/L after 21 days, extending the induction period for the osteogenic differentiation of PDLSCs. Drug release from carriers is influenced by various factors, including the drug-to-polymer ratio, the physical and chemical interactions between components, and the degradation rate of the polymeric scaffold. Typically, an initial burst release is observed.

PDLSCs are a type of human periodontal ligament-derived mesenchymal stem cell and are extremely suitable for osteogenesis-related research. Many studies have used these cells as a tool for bone differentiation research [26,48]. The culture of PDLSCs on PCL–ZnO nanofibers emphasizes that PCL fibers are non-toxic to PDLSCs and may promote slight proliferation. However, adding ZnONPs decreases cell viability, with viability decreasing as the ZnONP content increases, indicating that ZnONPs have a certain level of cytotoxicity. This study is consistent with the study by Nasajpour et al. [49], which attributed excessive production of H_2_O_2_ to the ZnO used during cell culture, leading to high levels of reactive oxygen species.

In many previous studies, biomaterials have typically used hydroxyapatite as an osteoinductive agent and ZnO as an antibacterial agent [15,50,51]. However, ZnONPs themselves possess significant osteoinductive capabilities. After culturing PDLSCs on PCL–ZnO nanofibers for seven days, ALP activity was significantly higher compared with culturing on pure PCL fibers, with an increase of approximately 56.9%. However, there was no significant difference in ALP expression between the 1% and 2% ZnO groups. The mineral deposition in the PCL–1% ZnO and PCL–2% ZnO groups was significantly higher compared with the PCL group at 21 days, with increases of 119% and 230%, respectively. During osteogenic differentiation throughout the early, mid, and late stages, the gene expression levels of Col-1, Runx2, BSP, and OCN in the PCL–ZnO group were 1.70- to 2.18-fold higher than those in the PCL group. For protein expression, the levels of Runx2 and OCN in the PCL–ZnO group were 30–50% higher than those in the PCL group. These results are consistent with the findings obtained by Tang et al. [2], Wang et al. [52], Seshadri et al. [53], and Harikrishnan et al. [54], who demonstrated the osteogenic differentiation effects of ZnO on MG-63 cells. Our study confirms that ZnONPs not only induce osteoblast-like MG-63 cells but also exhibit excellent efficacy in promoting the osteogenic differentiation of stem cells.

## 5. Conclusions

The optimal conditions for extracting EbM were determined by using RSM, resulting in a 1.79-fold increase in total flavonoid yield. By reacting EbME and zinc nitrate, 13 nm ZnONPs with a wurtzite structure were synthesized through a green biosynthesis approach. PCL–ZnO nanofibers provide a growth platform for PDLSCs similar to the natural extracellular matrix. The biodegradation of PCL releases Zn^2+^ ions, which induce the differentiation of PDLSCs into osteoblasts. This is evidenced by significant increases in ALP activity, mineral deposition, and the expression of early-, mid-, and late-stage osteogenic genes and proteins. These findings confirm that ZnONPs at low doses exhibit no cytotoxicity and are suitable for applications in regenerative medicine. To date, there are no clinical trials specifically involving ZnONPs, highlighting the need for further breakthroughs in human studies to validate their osteogenic therapy potential.

## Figures and Tables

**Figure 1 materials-18-02295-f001:**
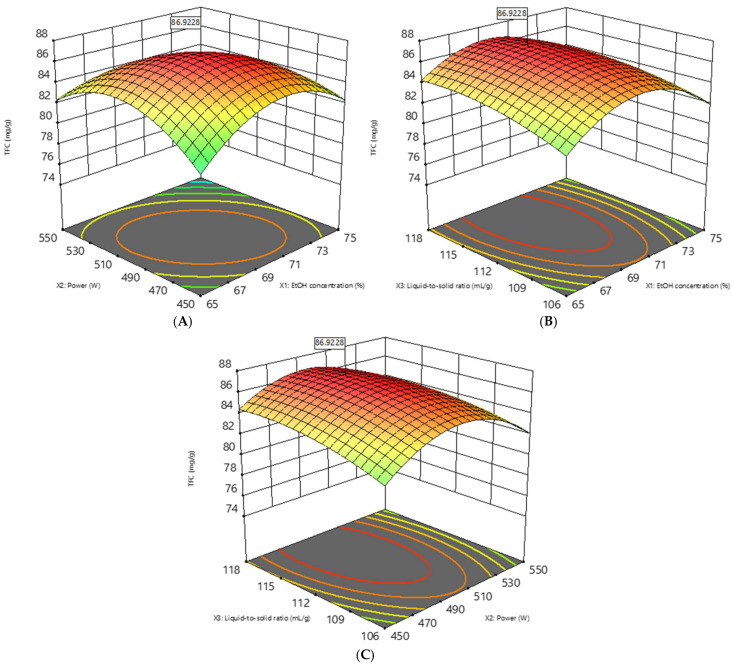
Three-dimensional response surface curve of the exact yield of EbM under MAE, showing the interactions between the ethanol concentration and power (**A**), ethanol concentration and solid/solvent ratio (**B**), and power and solid/solvent ratio (**C**).

**Figure 2 materials-18-02295-f002:**
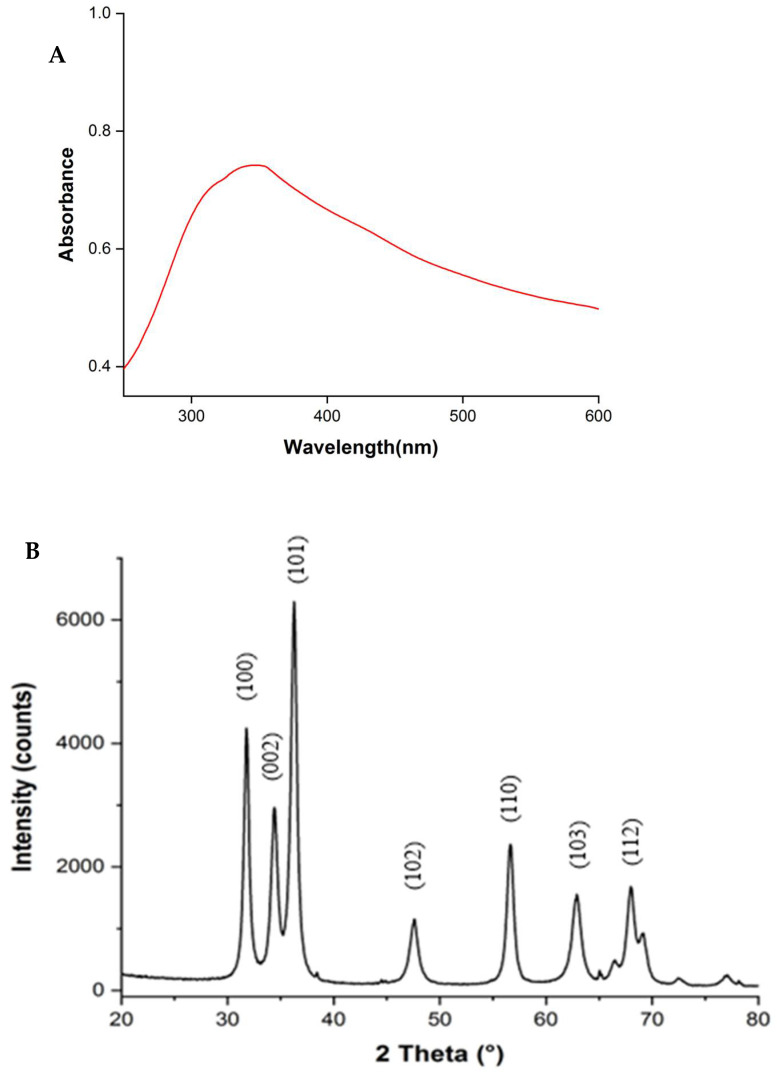
UV−Vis absorption spectrum (**A**), XRD pattern (**B**), FTIR spectra (**C**), SEM images (**D**), EDS analysis (**E**), TEM image (**F**), size distribution (**G**), SAED pattern (**H**), antibacterial ability (**I**), and zeta potential (**J**) of ZnONPs synthesized with EbME.

**Figure 3 materials-18-02295-f003:**
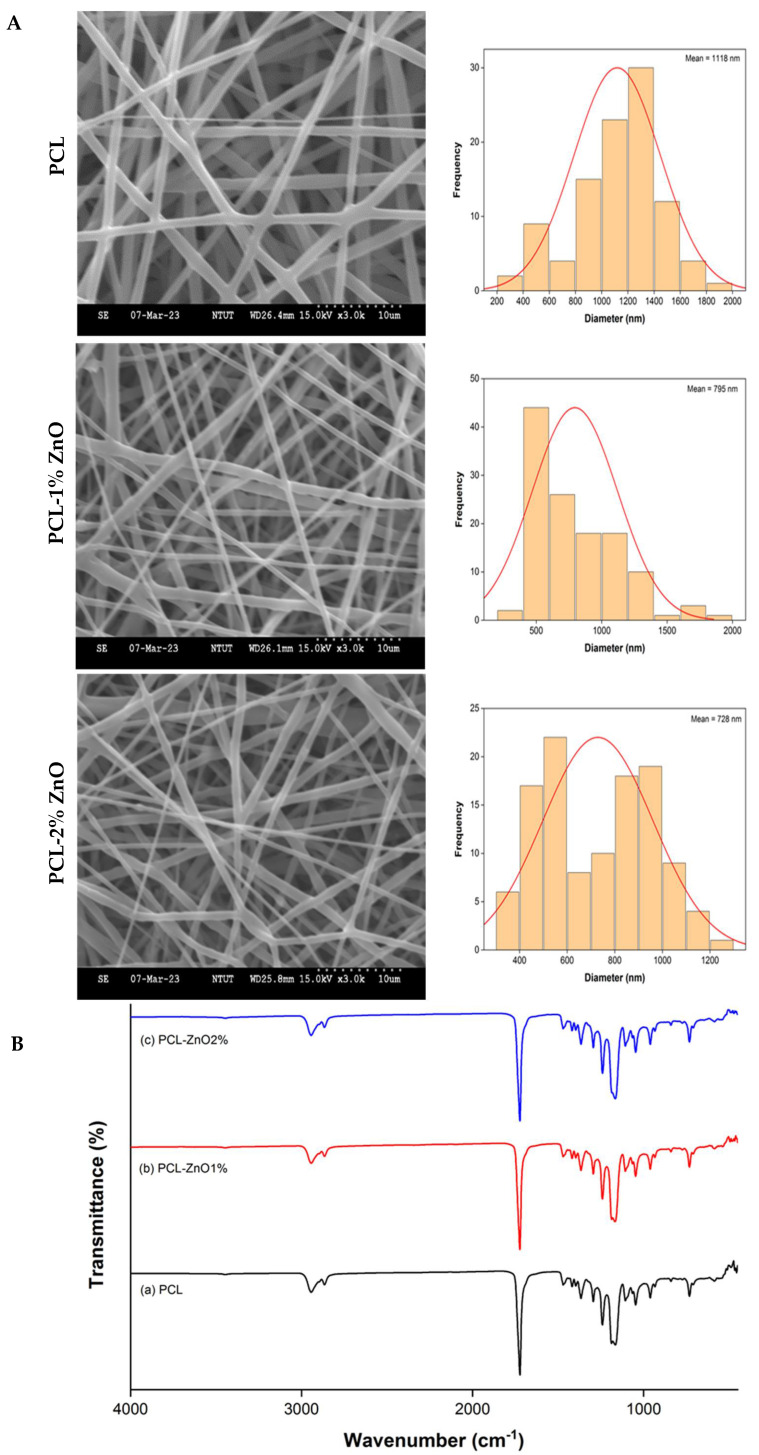
SEM images and diameter distribution (**A**), FTIR spectra (**B**), XRD pattern; (**C**), swelling capacity (**D**), water contact angles (**E**), mechanical property (**F**), degradation rate (**G**), Zn^2+^ ion release (**H**) of PCL−ZnONP nanofibers, and rheological properties of PCL–ZnONP solution (**I**). Values are expressed as mean ± SD (*n* = 3); * compared with PCL group: * *p* < 0.05, ** *p* < 0.01, and *** *p* < 0.001.

**Figure 4 materials-18-02295-f004:**
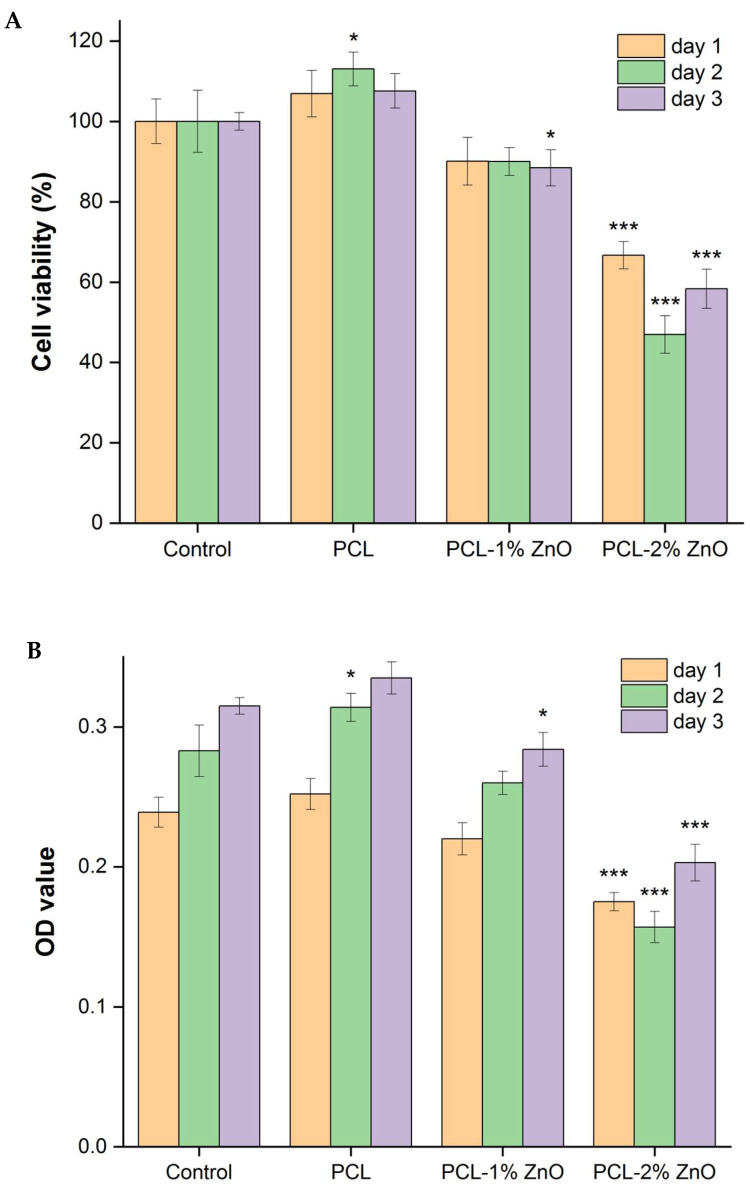
Viability and cytotoxicity (**A**,**B**), ALP activity (**C**), and calcification deposits by Alizarin Red S staining; magnification: ×10 (**D**) of PDLSCs treated with PCL–ZnO nanofibers. Values are expressed as means ± SD (*n* = 3); * compared with control or PCL group: * *p* < 0.05, ** *p* < 0.01, and *** *p* < 0.001.

**Figure 5 materials-18-02295-f005:**
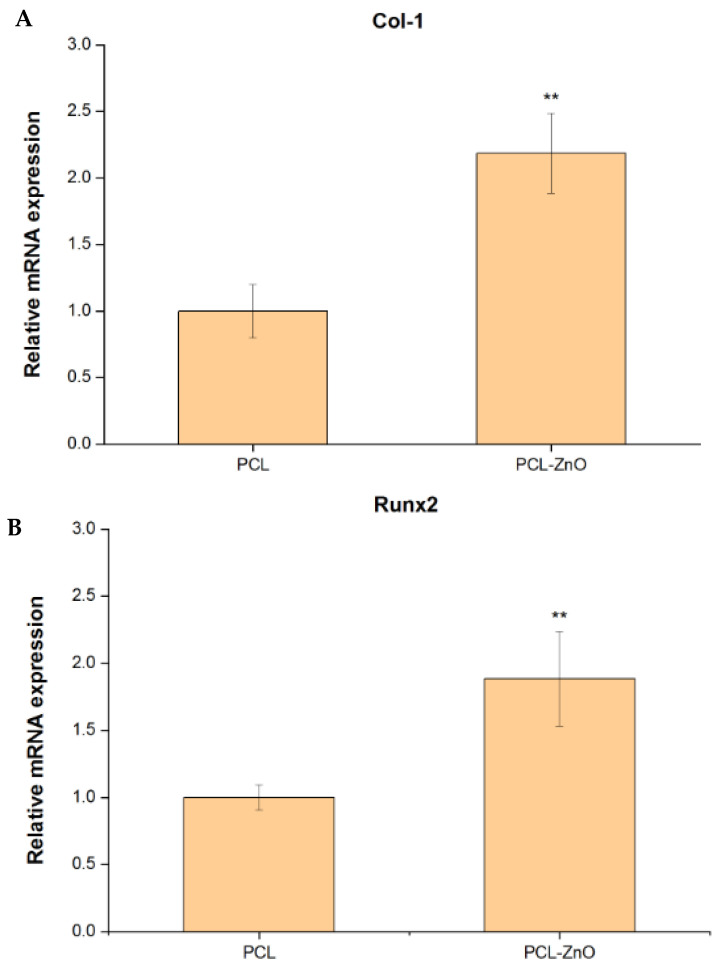
The quantitative RT-PCR gene expression levels of Col-1 (**A**), Runx-2 (**B**), BSP (**C**), and OCN (**D**) and the quantitative Western blotting protein expression levels of Runx-2 (**E**) and OCN (**F**) in PDLSCs treated with PCL–ZnO nanofibers for 7–21 days of osteogenic differentiation. Values are expressed as means ± SD (*n* = 3); * compared with PCL group or and # compared with 7 or 14 days of PCL–ZnO group: * *p* < 0.05, ** *p* < 0.01, and *** and ### *p* < 0.001.

**Table 1 materials-18-02295-t001:** Independent variables in a two-level factorial design.

Factor	Independent Variable	Code Level		
		−1	0	+1
X_1_	Time (min)	20	30	40
X_2_	EtOH concentration (%)	50	60	70
X_3_	Temperature (°C)	60	70	78
X_4_	Power (W)	500	600	700
X_5_	Liquid/solid ratio (mL/g)	80	100	120

**Table 2 materials-18-02295-t002:** The ethanol concentration (%), power (W), and solvent/solid ratio (mL/g) along the steepest ascent path for the extract of *Epimedium brevicornum* Maxim. in the response surface methodology at 30 min and 70 °C.

	X_1_ (min)	X_2_ (%)	X_3_ (°C)	X_4_ (W)	X_5_ (mL/g)
(1) Base point	30	60	70	600	100
(2) Unit		10		100	20
(3) Slope	−0.6964	4.97	5.23	−5.12	2.91
(4) =(2) × (3)		49.7		−512	58.2
(5) New unit = (4) × 0.11 *		5		−50	6
0	30	60	70	600	100
1		65		550	106
2		70		500	112
3		75		450	118

**Table 3 materials-18-02295-t003:** Central composite design for extract of *Epimedium brevicornum* Maxim. in response surface methodology.

Run	X_2_: EtOH Concentration	X_4_: Power	X_5_: Solvent/Solid Ratio	Total Flavonoids
	%	W	mL/g	mg/g QE
1	65	450	106	75.70 ± 1.25
2	75	450	106	79.97 ± 0.14
3	65	550	106	79.14 ± 0.15
4	75	550	106	76.47 ± 0.37
5	65	450	118	80.35 ± 0.64
6	75	450	118	83.20 ± 1.05
7	65	550	118	82.25 ± 0.53
8	75	550	118	77.37 ± 0.92
9	62	500	112	80.37 ± 0.81
10	78	500	112	75.59 ± 1.02
11	70	425	112	81.43 ± 0.09
12	70	575	112	78.94 ± 0.20
13	70	500	102	85.66 ± 1.03
14	70	500	122	83.82 ± 1.18
15	70	500	112	86.63 ± 0.47
16	70	500	112	86.61 ± 0.44
17	70	500	112	86.58 ± 0.95

## Data Availability

The original contributions presented in this study are included in the article/Appendix A. Further inquiries can be directed to the corresponding authors.

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
