# Peer review of "Incorporation of Zinc Oxide Nanoparticles Biosynthesized from Epimedium brevicornum Maxim. into PCL Nanofibers to Enhance Osteogenic Differentiation of Periodontal Ligament Stem Cells"

_materials, 2025, doi:10.3390/ma18102295_

Round 1

Reviewer 1 Report

Comments and Suggestions for Authors

Although the topic has no novelty in the field, however, this article should be improved as following:

1- SEM require mapping to confirm distribution of Zn.

2- It would be higher quality paper if authors measure the TGA of samples.

3- As long as Zn has antimicrobial characteristics, required test must be performed on samples.

Author Response

Response to Reviewers

The authors would like to express my sincerely thanks to the editor and reviewers for the professional comments and constructive suggests. The article was revised following the suggestions. The modifications are listed as below.

Reviewer#1:

  1. SEM require mapping to confirm distribution of Zn.

Response: The distribution of ZnO NPs has been annotated on the SEM images.

  1. It would be higher quality paper if authors measure the TGA of samples.

Response: TGA can illustrate the physical and chemical properties of materials and is an important reference data. However, PCL/ZnO nanofiber, as a cell culture scaffold at 37℃, will not face high-temperature lysis. PCL is a biodegradable polymer, and ZnO will dissolve and release zinc ions as an inducer of bone differentiation. However, in order to fully characterize the material, TGA testing will be included in our future studies.

  1. As long as Zn has antimicrobial characteristics, required test must be performed on samples.

Response: We have included the antibacterial data of ZnO in the results section.

Reviewer 2 Report

Comments and Suggestions for Authors

The manuscript entitled, ‘Incorporation of zinc oxide nanoparticles biosynthesized from Epimedium brevicornum Maxim. into PCL nanofibers to enhance osteogenic differentiation of periodontal ligament stem cells’ reported Zinc oxide based polymer composites for biomedical applications. The article should be modified according the following comments:

  1. The abstract lacks specificity regarding the data presented in the study. It is recommended to highlight key findings or notable data points to give readers a clearer understanding of the study's contributions.
  2. The XRD peaks are assigned based on JCPDS No. 36-1451, confirming the wurtzite structure of ZnONPs. Did you analyze the crystallinity percentage or provide any Rietveld refinement to further support phase purity and structural confirmation?
  3. ZnO nanoparticles can sometimes exhibit secondary phases or defects. Did you perform any additional techniques, such as Raman spectroscopy or photoluminescence (PL) studies, to investigate defects or oxygen vacancies in the ZnONPs?
  4. While ZnONPs exhibited cytotoxicity, they also enhanced ALP activity and mineralization. Can you provide insights into the optimal concentration range where ZnONPs maximize osteogenic differentiation without compromising cell viability?
  5. Alizarin Red S staining results suggest a dose-dependent effect of ZnONPs. Did you perform calcium quantification assays or SEM-EDX analysis to validate and quantify mineral deposition more precisely?
  6. Since ZnONPs can exhibit ion release and degradation over time, did you analyze the stability of the PCL-ZnONP scaffolds in culture medium to confirm their sustained bioactivity?
  7. Runx2 and OCN expression levels were analyzed at both mRNA and protein levels. Did you perform correlation analysis to assess the consistency between gene transcription and protein expression trends?
  8. The study assesses gene and protein expression up to 21 days. Did you evaluate the potential long-term effects of ZnONPs on osteogenic maturation and mineralization beyond this time frame?
  9. Some articles would be significance for your reference:
  • Scherer, F., Wille, S., Saure, L., Schütt, F., Wellhäußer, B., Adelung, R., & Kern, M. (2024). Investigation of Mechanical Properties of Polymer-Infiltrated Tetrapodal Zinc Oxide in Different Variants. Materials17(9), 2112.
  • Ganguly, S., & Margel, S. (2023). Superhydrophobic nanoscale materials for surface coatings. In Polymer-Based Nanoscale Materials for Surface Coatings(pp. 479-500). Elsevier.

Author Response

Response to Reviewers

The authors would like to express my sincerely thanks to the editor and reviewers for the professional comments and constructive suggests. The article was revised following the suggestions. The modifications are listed as below.

Reviewer#2:

  1. The abstract lacks specificity regarding the data presented in the study. It is recommended to highlight key findings or notable data points to give readers a clearer understanding of the study's contributions.

Response: We have added some notable data in the Abstract.

  1. The XRD peaks are assigned based on JCPDS No. 36-1451, confirming the wurtzite structure of ZnONPs. Did you analyze the crystallinity percentage or provide any Rietveld refinement to further support phase purity and structural confirmation?

Response: In this study, ZnONPs were used as an inducer of osteogenic differentiation for PDLSC, and no in-depth analysis of the structure of ZnONPs was performed.

  1. ZnO nanoparticles can sometimes exhibit secondary phases or defects. Did you perform any additional techniques, such as Raman spectroscopy or photoluminescence (PL) studies, to investigate defects or oxygen vacancies in the ZnONPs?

Response: As in the answer to question 2, in future research, we will conduct a complete analysis of the material based on the reviewers' comments.

  1. While ZnONPs exhibited cytotoxicity, they also enhanced ALP activity and mineralization. Can you provide insights into the optimal concentration range where ZnONPs maximize osteogenic differentiation without compromising cell viability?

Response: The cytotoxicity of metal nanoparticles or metal oxides is affected by many factors, including particle size, synthesis method, cell type, dose/concentration, exposure time, and administration route. Stem cell differentiation is generally induced at a dose that allows for maximum cytotoxicity. Based on the comprehensive results of cell viability, ALP activity and mineralization analysis, 1% ZnO was finally selected as the induction dose for osteogenic differentiation gene and protein expression.

  1. Alizarin Red S staining results suggest a dose-dependent effect of ZnONPs. Did you perform calcium quantification assays or SEM-EDX analysis to validate and quantify mineral deposition more precisely?

Response: Unfortunately, we did not perform SEM-EDX analysis on the induced deposited minerals. Because ARS staining is obtained by reaction to calcium ions, it is the most important technique for general bone differentiation analysis.  

  1. Since ZnONPs can exhibit ion release and degradation over time, did you analyze the stability of the PCL-ZnONP scaffolds in culture medium to confirm their sustained bioactivity?

Response: PCL is a biodegradable polymer, so PCL-ZnONP fiber can be continuously degraded in the culture medium and continuously release zinc ions to induce differentiation of stem cells because PCL is a biodegradable polymer. Figure 3G and 3H are the results of PCL-ZnONP fiber degradation and zinc ion release.

  1. Runx2 and OCN expression levels were analyzed at both mRNA and protein levels. Did you perform correlation analysis to assess the consistency between gene transcription and protein expression trends?

Response: In Figure 5, Runx2 and OCN show a positive correlation between gene and protein expression, indicating that osteogenic differentiation of stem cells continues, but the expression folds of genes and proteins at the same time may be slightly different.

  1. The study assesses gene and protein expression up to 21 days. Did you evaluate the potential long-term effects of ZnONPs on osteogenic maturation and mineralization beyond this time frame?

Response: Unfortunately, the potential long-term effects of ZnONPs on osteogenic maturation and mineralization were not evaluated after 21 days of culture. However, this view deserves more detailed follow-up evaluation in future studies.

  1. Some articles would be significance for your reference:

Response: Thanks to the reviewer’s suggestion, we have added the article of Scherer et al. to the references 49, and the article of Ganguly et al. is not suitable for reference in this study.

Reviewer 3 Report

Comments and Suggestions for Authors

On request of Materials, I have revised the manuscript titled “Incorporation of zinc oxide nanoparticles biosynthesized from Epimedium brevicornum Maxim. into PCL nanofibers to enhance osteogenic differentiation of periodontal ligament stem cells”, by Kuei-Ping Hsiehet et al.

Here, the Authors have proposed an eco-friendly method to prepare ZnO NPs from zinc nitrate, using a microwave-assisted extract of Epimedium brevicornum Maxim. The following incorporation of ZnO NPs into polycaprolactone (PCL) provided highly hydrophilic composite fibres with increased Young’s modulus, biodegradation rate, swelling ratio, while decreased diameter and water contact angle. Interesting, the prepared ZnO/PCL fibres demonstrated excellent biocompatibility towards PDLSCs and effectively induced their osteogenic differentiation.

General comments

As for the synthetic technique used by the Authors to prepare ZnO NPs, this work represents yet another variant of the often-described green method to prepare metal nanoparticles. However, the work is well written, and its design is rational. The results are promising, and the prepared nanofibers could be developed in the future as materials to effectively promote PDLSCs osteogenic differentiation.

Anyway, some major and minor revision are needed to improve further the quality of this work.

Authors should determine the dimensional distribution of ZnO NPs and of ZnO-PCL composite fibres using also DLS analyses, which will allow to determine PDI and Zeta potential of particle, as well.

Authors should determine the equilibrium swelling rate (%) by measuring the swelling rate over time at least in triplicate and provide the related graph.

Rheological characteristic of PCL-ZnO fibres should be investigated. Graphs of viscosity vs shear rate and of shear stress vs shear rate should be provided to demonstrate the rheological behaviour of fibres (Bigham plastic, pseudoplastic, shear thinning, shear thickening, etc).

Error bars are missing in graph in Figure 3H. Please, provide them.

Lines 621-622. This conclusion is rather risky. Do the Authors know how long the process leading to the therapeutic application of a compound is? It certainly cannot be asserted that a material that has only been tested in vitro can be used in therapy. Many in vivo experiments and clinical trials are necessary before one can even think of clinically applying the ZnO-PCL nanofibers proposed by the Authors.

Minor

Lines 47, 338, 473, 585 and 618. Please, 2+ should be superscript.

Line 174. Please, change “cellular” with “cell”.

Table x. and Figure x. in bold.

The title of Tables and captions of Figures without indent.

In the text refer to Figures using “Figure” and not Fig.

On these considerations I ask Authors major revisions.

Author Response

Response to Reviewers

The authors would like to express my sincerely thanks to the editor and reviewers for the professional comments and constructive suggests. The article was revised following the suggestions. The modifications are listed as below.

Reviewer#3:

  1. Authors should determine the dimensional distribution of ZnO NPs and of ZnO-PCL composite fibres using also DLS analyses, which will allow to determine PDI and Zeta potential of particle, as well.

Response: The authors once used DLS to analyze the PDI and Zeta potential of ZnO NPs, but the data was not ideal, which may be because the ZnO NPs aggregated and could not be dispersed, so it was not presented in the results. The size distribution of ZnO-PCL composite fibers is generally presented by SEM.

  1. Authors should determine the equilibrium swelling rate (%) by measuring the swelling rate over time at least in triplicate and provide the related graph.

Response: In Figure 3D had presented the swelling capacity of PCL and PCL/ ZnONPs. After soaking for 1 day, the swelling rate is about 200~300% at least in triplicate .The swelling rate over time is not measured, because the fiber will degrade if the soaking time is too long.

  1. Rheological characteristic of PCL-ZnO fibres should be investigated. Graphs of viscosity vs shear rate and of shear stress vs shear rate should be provided to demonstrate the rheological behaviour of fibres (Bigham plastic, pseudoplastic, shear thinning, shear thickening, etc).

Response: In this study, ZnONPs were used as an inducer of PDLSC bone differentiation. PCL/ZnONPs fiber is a 3D platform for culturing cells, and been made by electrospinning technology. The research process did not involve the use of PCL/ZnONPs flow or colloid, so there is no need to perform rheological characteristic analysis.

  1. Error bars are missing in graph in Figure 3H. Please, provide them.

Response: We have added the error bar in the Figure H. Thanks reviewer’s suggestion.

  1. Lines 621-622. This conclusion is rather risky. Do the Authors know how long the process leading to the therapeutic application of a compound is? It certainly cannot be asserted that a material that has only been tested in vitro can be used in therapy. Many in vivo experiments and clinical trials are necessary before one can even think of clinically applying the ZnO-PCL nanofibers proposed by the Authors.

Response: Thank you very much, we have revised it to a more conservative statement.

  1. Minor

Lines 47, 338, 473, 585 and 618. Please, 2+ should be superscript.

Line 174. Please, change “cellular” with “cell”.

Table x. and Figure x. in bold.

The title of Tables and captions of Figures without indent.

In the text refer to Figures using “Figure” and not Fig.

Response: Thank you very much. We have carefully corrected the reviewer's suggestions.

Round 2

Reviewer 1 Report

Comments and Suggestions for Authors

It is acceptable now.

Author Response

We are truly grateful for the reviewer’s positive comments and recognition.

Reviewer 2 Report

Comments and Suggestions for Authors

The author did most of the comments. So this can be accepted. 

Author Response

(The authors gave the same response as above.)

Reviewer 3 Report

Comments and Suggestions for Authors

Dear Authors,

responding to my first comment, you explain that it was not possible carry out DLS analysis, because your NPs are poorly disperible in water. Not do you thing that this characteristic can be a great problem in sight of in vivo administrations. This fact strongly lower the scientific sound of this work.

Responding to my second comment, you explain that you did not carry out swelling experiments over time because your NPs degrade for suspension in water longer than 1 day. Anyway, I have not asked to go over this time, but to make swelling measurements at fixed points until the equilibrium is reached, which could be a time < 1 day. Please, perform these experiments.

Point 3. I apologise in advance, but I disagree with you. Rheolgical experiments I asked are necessary for a swelling materials.

Author Response

Response to Reviewers

The authors would like to express my sincerely thanks to the editor and reviewers for the professional comments and constructive suggests. The article was revised following the suggestions. The modifications are listed as below.

Reviewer#3:

  1. responding to my first comment, you explain that it was not possible carry out DLS analysis, because your NPs are poorly disperible in water. Not do you thing that this characteristic can be a great problem in sight of in vivo administrations. This fact strongly lower the scientific sound of this work.

Response: We regret that the reviewer may have misunderstood our previous response. The authors did perform DLS analysis on ZnO nanoparticles, and the measured particle sizes were 434.3 nm and 402.8 nm. The data are as follows. However, these results were clearly inconsistent with the TEM observations. Therefore, we did not include the DLS data in the manuscript. As mentioned in our initial response, "The authors once used DLS to analyze the PDI and Zeta potential of ZnO NPs, but the data was not ideal, which may be because the ZnO NPs aggregated and could not be dispersed, so it was not presented in the results. The size distribution of ZnO-PCL composite fibers is generally presented by TEM."

  1. Responding to my second comment, you explain that you did not carry out swelling experiments over time because your NPs degrade for suspension in water longer than 1 day. Anyway, I have not asked to go over this time, but to make swelling measurements at fixed points until the equilibrium is reached, which could be a time < 1 day. Please, perform these experiments.

Response: We have added the data of swelling within 1 day, as reviewer’s suggestion.

  3. I apologise in advance, but I disagree with you. Rheolgical experiments I asked are necessary for a swelling materials.

Response: The authors have analyzed the rheological characteristics of PCL-ZnO solutions, as reviewer’s suggestion. The shear stress vs shear rate graph passes through the origin and shows a concave downward as the shear rate increases, confirming that PCL and PCL-ZnO are pseudoplastic fluids. However, the viscosity of PCL and PCL-ZnO solutions showed a decreasing trend with the increase of shear rate.

The figures was shown in attach.

Round 3

Reviewer 3 Report

Comments and Suggestions for Authors

Dear Authors,

I have understand your explanation about the DLS analyses. Anyway, the dimentional distributions obtained by DLS, with the explanation you have provided me, should be inserted in the main text. Comments on the PDI value should be provided. The DLS analysis to assess the Z-potential values should be performed and the results included in the text. 

An experimental part and a disccussion, for new rhelogical experiments carried out and showed me, must be inserted in the manuscript, together with the related graphs. Sincerly, the graphs of shear stress versus shear rate leave me a bit perplexed. I am used to seeing different ones (see Figure below) for pseudoplastic materials and they do not seem to agree with the graphs of viscosity versus shear rate, which are instead characteristic of pseudoplastic materials. Therefore, a discussion/explanation on this fact is necessary. Additionally, you should define if your materials are shear thinnin or shear thikening and give the values of yead stress.

Also, by reconsidering your work, I noticed that no mathematical kinetic model has been used to process the Zn2+ ion release profiles in panel H of Figure 3. This operation is important to establish the kinetics/mechanisms which governe the ion release. Please, apply to the release data at least five of most used mathematical models. For example, consider https://doi.org/10.3390/nano10061243.   

Newtonian and Non-Newtonian Fluids | Newton's Law of Viscosity

Author Response

Response to Reviewers

The authors would like to express my sincerely thanks to the editor and reviewers for the professional comments and constructive suggests. The article was revised following the suggestions. The modifications are listed as below.

Reviewer#3:

  1. I have understaned your explanation about the DLS analyses. Anyway, the dimentional distributions obtained by DLS, with the explanation you have provided me, should be inserted in the main text. Comments on the PDI value should be provided. The DLS analysis to assess the Z-potential values should be performed and the results included in the text.

Response: The authors used DLS to analyze the ZnO nanoparticle size, which was about 430nm, with a PDI of 0.596 and a zeta potential of -23.73mV (Figure 2J). This result has been added to the manuscript.

  1. An experimental part and a discussion, for new rhelogical experiments carried out and showed me, must be inserted in the manuscript, together with the related graphs. Sincerly, the graphs of shear stress versus shear rate leave me a bit perplexed. I am used to seeing different ones (see Figure below) for pseudoplastic materials and they do not seem to agree with the graphs of viscosity versus shear rate, which are instead characteristic of pseudoplastic materials. Therefore, a discussion/explanation on this fact is necessary. Additionally, you should define if your materials are shear thinnin or shear thikening and give the values of yead stress.

Response: We have added the rheological properties in the manuscript, as reviewer’s suggestion.

  1. That no mathematical kinetic model has been used to process the Zn2+ ion release profiles in panel H of Figure 3. This operation is important to establish the kinetics/mechanisms which governe the ion release. Please, apply to the release data at least five of most used mathematical models. For example, consider https://doi.org/10.3390/nano10061243. 

Response: According to the reviewer's suggestion, the zinc ion release data were substituted into the zero-order, first-order, Higuchi, Hixoson-Crowel and Korsmeyer-Peppas models for kinetic analysis, and it was inferred that it belonged to the first-order kinetic release model (R2 was the largest). However, because the results are not perfect, they are only described in words in the results and discussion of the manuscript. The analysis results can be found in the Figure S1 (Supplementary Materials).
